# Multimorbidity and health system priorities in Zimbabwe: A participatory ethnographic study

Justin Dixon[1,2,3]*, Efison Dhodho[2,3☙], Fionah Mundoga[1,2☙], Karen Webb[2☙], Pugie Chimberengwa[2,4], Trudy Mhlanga[2], Tatenda Nhapi[2], Theonevus T. Chinyanga[2], Justice Mudavanhu[5], Lee Nkala[5], Ronald Nyabereka[6], Gwati Gwati[7], Gerald Shambira[8], Trust Zaranyika[9], Clare I. R. Chandler[3‡☙], Rashida A. Ferrand[1,10‡☙], Chiratidzo E. Ndhlovu[9‡☙]

1 The Health Research Unit Zimbabwe, Biomedical Research and Training Institute, Harare, Zimbabwe, 2 Organisation for Public Health Interventions and Development, Harare, Zimbabwe, 3 Department of Global Health and Development, London School of Hygiene and Tropical Medicine, London, United Kingdom, 4 National University of Science and Technology, Bulawayo, Zimbabwe, 5 Directorate of Non-Communicable Diseases, Ministry of Health and Child Care, Harare, Zimbabwe, 6 AIDS and TB Unit, Ministry of Health and Child Care, Harare, Zimbabwe, 7 Directorate of Policy, Planning, and Health Economics, Ministry of Health and Child Care, Harare, Zimbabwe, 8 Department of Community Medicine, Faculty of Medicine and Health Sciences, University of Zimbabwe, Harare, Zimbabwe, 9 Department of Internal Medicine, Faculty of Medicine and Health Sciences, University of Zimbabwe, Harare, Zimbabwe, 10 Department of Clinical Research, London School of Hygiene and Tropical Medicine, London, United Kingdom

☙ These authors contributed equally as second authors.
‡ CC, RF, and CN also contributed equally as last authors.
* justin.dixon@lshtm.ac.uk

## Abstract

Multimorbidity, increasingly recognised as a global health challenge, has recently emerged on the health agendas of many countries experiencing rapid epidemiological change, including in Africa. Yet with its conceptual origins in the global North, its meaning and possible utility in African contexts remains abstract. This study drew together policymakers, public health practitioners, academics, health informaticians, health professionals, and people living with multimorbidity (PLWMM) in Zimbabwe to understand: What is the transformative potential and possible limitations of elevating multimorbidity as a priority in this setting? To bring these different perspectives into conversation, we used a participatory ethnographic design that involved a health facility survey, participant-observation, in-depth interviews, audio-visual diaries, and participatory workshops. We found that multimorbidity was new to many respondents but generally viewed as a meaningful and useful concept. It captured the increasingly complex health profile of Zimbabwe's ageing population, foregrounded a range of challenges related to the 'vertical' organisation and uneven funding of different conditions, and revealed opportunities for integration across entrenched silos of knowledge and practice. However, with capacity and momentum to address multimorbidity concentrated within the HIV programme, there was concern that multimorbidity could itself become verticalized, undercutting its transformative potential. Participants agreed that responding to multimorbidity requires a decisive shift from

**Data availability statement:** Data from the study has been deposited with the United Kingdom Data Service (UKDS). We have included all transcripts (in-depth interviews), health facility survey data, and second-order summaries of fieldnotes from observational work. In line with our ethical approval, we have excluded raw fieldnotes, workshop discussions, and audio-visual diaries due to the challenges in maintaining confidentiality. The DOI for the data in UKDS is: https://doi.org/10.5255/UKDA-SN-857310.

**Funding:** This research was funded by a Wellcome Trust Research Fellowship in Humanities and Social Sciences (Multimorbidity and Knowledge Architectures: An Interdisciplinary Global Health Collaboration, ref. 222177, to JD, who received salary from this grant). The research was embedded within OPHID Target, Accelerate and Sustain Quality Care (TASQC) for HIV Epidemic Control (ref. 72061320CA00005) with support from the President's Emergency Plan for AIDS Relief (PEPFAR) through USAID (to ED, KW, PC, TM, TN and TTC, all of whom received salaries from the TAQC grant). The funders had no role in study design, data collection and analysis, decision to publish, or preparation of the manuscript.

**Competing interests:** The authors have declared that no competing interests exist.

vertical, disease-centred programming to restore the comprehensive primary care that undergirded Zimbabwe's once-renowned health system. It also means building a policy-enabling environment that values generalist (as well as specialist) knowledge, ground-level experience, and inclusive stakeholder engagement. We conclude that the 'learning' health system represents a promising conceptual lens for unifying these imperatives, providing a tangible framework for how knowledge, policy, and practice synergise within more self-reliant, person-centred health systems able to respond to complex health challenges like multimorbidity.

## 1. Introduction

Multimorbidity, commonly defined as the co-occurrence of two-or-more long-term conditions in one individual, has been described as the next 'global pandemic' [1]. In countries undergoing epidemiological transitions, multimorbidity has been characterised as a 'clash' of persisting communicable diseases (notably HIV and tuberculosis (TB)) and rising non-communicable diseases (NCDs) that require multiple forms of expertise and coordinated care to manage [2]. Yet, health systems remain largely organised around specialist rather than generalist knowledge [3]. In many African nations, this translates into siloed organisation of care, fuelled by 'vertical' single-disease programming [2]. Priority-setting initiatives for multimorbidity in a global context [4–6] including in sub-Saharan Africa [5,6] highlight the need to identify common disease 'clusters' and determinants; improve multimorbidity prevention and treatment; and more broadly restructure health systems to become more integrated and person-centred. The COVID-19 pandemic has since highlighted the importance of prioritising multimorbidity, with the virus disproportionately affecting those with 'underlying conditions' [7].

The social sciences, including medical anthropology, have been integral in advancing understandings of multimorbidity. Qualitative studies in low-resource contexts including in Africa have identified systemic vulnerabilities that expose marginalised populations to multimorbidity [8,9], how health and social challenges interact to produce and exacerbate multimorbidity [10,11], and the challenges faced by patients and providers navigating fragmented health systems [12–15]. Qualitative studies have also contributed to recognition that frameworks to integrate care are extremely challenging to implement and often fall short of tackling the social and structural determinants of multimorbidity. For instance, Bosire et al. [12] demonstrated that South Africa's Integrated Chronic Disease Management (ICDM) Model, designed to provide person-centred care for long-term conditions, faced considerable challenges delivering on its intended outcomes. This was due to resource scarcity, partial implementation, continued fragmentation, and emphasis on pharmaceutical care at the expense of patients' social and economic context.

With multimorbidity gaining traction as a global health priority, scholars have examined what it is 'doing' as a concept and whether it is proving as radical and transformative as hoped [16–18]. On the one hand, multimorbidity has become a

focal point for tensions that have long been building within medicine and global health. It lays bare the evident limitations of 'vertical' funding and governance mechanisms, parallel research and data infrastructures, proliferating clinical (sub-) specialities and neglect of generalism and public health, and fragmented care delivery systems [17]. On the other hand, multimorbidity is a disease-centred concept and emergent of the Euro-Western philosophical traditions that made the single disease model so intractable [16–18]. Recognition of this has prompted several initiatives – including within the current Collection – to reconceptualise multimorbidity to put people, rather than diseases, back at its theoretical centre [16,19,20]. Yet with little empirical research into the meaning and prioritisation of multimorbidity in particular health system contexts, these conversations remain abstract. Inclusive, bottom-up understandings of multimorbidity are needed to ensure that multimorbidity is able to fulfil the promises that have been pinned to it and that it does not inadvertently perpetuate disease-centred care [17].

This article presents findings from a participatory ethnographic study that asked: what is the transformative potential and possible limitations of elevating multimorbidity as a priority in Zimbabwe? To answer this question, we brought together a diversity of perspectives on multimorbidity from across the country's health and academic sectors to better understand the tensions, challenges, and opportunities that multimorbidity brought to the fore and to identify with participants what it might mean to prioritise multimorbidity in this health system context. Because multimorbidity in Zimbabwe was only just starting to emerge on the radar (as in many African nations), this was an opportune moment to develop a formative agenda and set of priorities with participants while creating new partnerships and relationships to take these forward. The aim was to engage stakeholders in a way that was inclusive, critical, and facilitated engagement across different disciplines, fields, and perspectives. We thereby hoped to ensure that the transformative potentials of multimorbidity were optimised while minimising any possible limitations or harms.

## 2. Study setting and design

This article is based on findings from the KnowM^M study (2021–2024), an interdisciplinary global health research collaboration centred on a participatory ethnographic study of multimorbidity within the Zimbabwean health system.

### 2.1. Study setting

Zimbabwe is a lower-middle-income country with a population of 16.3 million [21]. Following independence in 1980, huge strides were made in expanding access to healthcare. This included moving from an urban, curative, and racially-biased health system to one focused on delivering affordable primary healthcare to underserved communities [22–24]. Zimbabwe came to boast one of Africa's strongest health systems, with thriving teaching hospitals and academic sector, well-trained health workforce, and decentralised care system organised around its essential medicine list, national treatment guideline (EDLIZ) and co-produced training manuals [25]. However, the achievements of the 1980s–90s were undone by political instability, economic structural adjustment (which decreased public spending in favour of privatisation), hyperinflation, the reintroduction of user fees in hospitals and urban clinics, and the HIV and AIDS epidemic [26]. Zimbabwe has since experienced among the steepest rises in NCDs in sub-Saharan Africa [27], with an estimated 40% and 13% adult prevalence of hypertension and diabetes, respectively [28]. Modelling also suggests that multimorbidity, particularly among the 12.9% of people in living with HIV, will rise sharply by 2035 [29]. The health system remains fragmented, under-resourced and oriented towards infectious diseases. It has also experienced high rates of health worker attrition and regular collapses, most recently during the COVID-19 pandemic [30].

KnowM^M was conducted in four provinces of Zimbabwe: Harare, Bulawayo, Mashonaland East, and Matabeleland South (Fig 1). Harare and Bulawayo are metropolitan provinces, with the majority of political institutions, universities, and all Central hospitals. They also both share a similar provincial health system structure, with primary healthcare mostly provided by City Councils and most referrals going straight to one of the Central hospitals. Both also have large private sectors made up of primary and secondary facilities (for the affluent minority) and private pharmacies and laboratories.

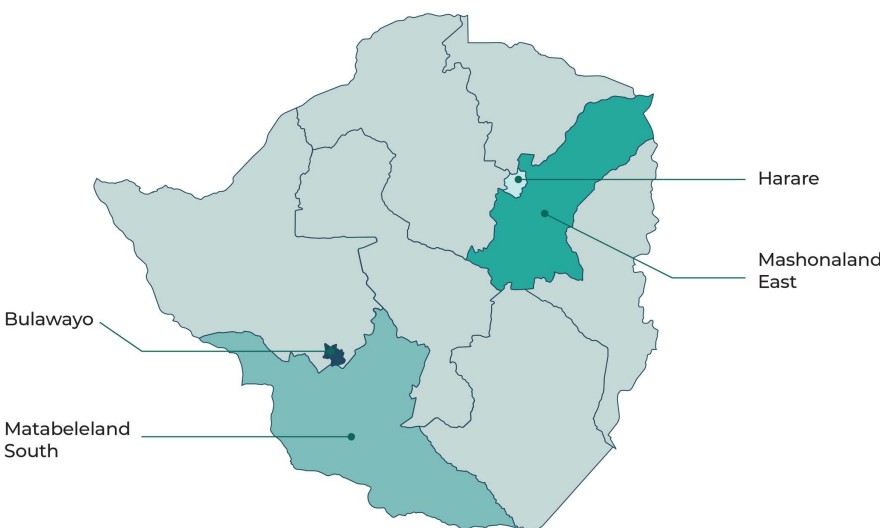

**Fig 1. Provinces of Zimbabwe included in the KnowM^M study\*.** \*Base map produced using shape files obtained from the Humanitarian Data Exchange (https://data.humdata.org/dataset/cod-ab-zwe).

Mashonaland East, bordering Harare, is a predominantly rural province in which most public healthcare is run by the central government, which includes primary, secondary, and tertiary care, with referrals for quaternary care generally sent to Harare. Matabeleland South is a largely rural province with a similar structure to Mashonaland East, with quaternary referrals mainly to Bulawayo rather than Harare. The four provinces were selected to capture urban and rural settings, provinces in both the majority Mashonaland and minority Matabeleland regions, and to capture the main differences in health system structure and referral relationships.

## 2.2. Study design

KnowM^M used a participatory ethnographic study design. The design was informed by formative work to map the emerging field of global multimorbidity research [6,17] and engagement in Zimbabwe with the Ministry of Health and Child Care (MoHCC). Primary data were collected between 01-09-2022 and 31-12-2023 by an interdisciplinary team of medical anthropologists (JD, FM, CIRC), clinical academics (RAF, GS, TZ, CEN), and public health practitioners from project implementing partner OPHID (KW, ED, PC, TM, TN, TTC), supported by the MoHCC (RN, JM, LN, GG). Our study design is informed by the principle of "slow co-production" [31] which includes participants not just as sources of data but as active agents in the production of knowledge. This was important since the concept of multimorbidity has emerged from high-income settings, with uncertainty about its relevance beyond the rich world. In engaging with participants, we invited them to think 'with' the concept of multimorbidity. Where appropriate, we provided them with the working definition of multimorbidity as 'two-or-more long-term conditions', but were careful during conversations not to reify it as a category and to allow participants to lead conversations about its meaning and significance in the Zimbabwean context.

130 individuals took part in the study (overview of participants provided in Table 1). Data collection commenced with a survey of health facilities (n = 30), purposively selected to provide an initial understanding of the capacity of facilities at different levels of care for addressing NCDs and multimorbidity (see Table 2) and to generate interest for follow-up field-work. For logistical reasons, all facilities were in the public and not-for-profit sectors; perspectives from the private sector were fed in through other methods. Quantitative and qualitative survey questions were developed based on the WHO Service and Readiness Assessment (SARA) survey [32], knowledge within our team on NCDs and multimorbidity, and

Table 1. Participant demographics and activities breakdown.

| Participant group | Ethnographic fieldwork participants (total and specific activities*) | | | | | Workshop participants | | Grand total participants |
|---|---|---|---|---|---|---|---|---|
| | Total participants | Facility survey | In-depth interviewees | Participant-observation | Audio-visual diaries | Bulawayo workshop | Harare workshop | |
| People living with multimorbidity | **23** | – | 23 | – | 10 | 2(2†) | 3(3) | **23** |
| Healthcare professionals# | **46** | 31‡ | 6 | 18 | – | 9(3) | 7(3) | **56** |
| Clinical academics and educators#) | **7** | | 5 | 7 | – | 1(1) | 4(2) | **9** |
| Public health policymakers and officers within the Ministry of Health and Child Care (MoHCC) | **5(7)‡** | – | 4 | 5 | – | 7(3) | 13(3) | **19** |
| Health informaticians, electronic health record (EHR) experts and/or data managers | **2(3)¶** | – | 3 | 2 | – | 1(0) | 1(0) | **4** |
| Non-governmental organization (NGOs) employees | **10(11)&** | – | 4 | 6 | – | 12(7) | 9(5) | **19** |
| **Cumulative total** | **93** | **31** | **45** | **38** | **10** | **32(16)** | **37(16)** | **130** |

*Some participants took part in multiple research activities.

#Fields and specialties represented by healthcare professionals and clinical academics included: general practice (GPs), general nursing, midwifery, infectious disease (mostly HIV and TB), rheumatology, endocrinology, oncology, psychiatry, mental health, epidemiology, and public health.

†Number of which were prior research participants in ethnography and excluded from the cumulative totals of participants.

‡In one central hospital, there were 2x survey respondents due to the semi-autonomous nature of its psychiatric unit but have merged these in the analysis to total n = 30 surveys.

‡n = 2 policymakers/officers co-classified as healthcare professionals and not added to cumulative total participants.

¶n = 1 health informatician officer co-classified as a policymaker and not added to cumulative total participants.

&n = 1 NGO employee co-classified an academic and not added to cumulative total participants.

Table 2. Health facility survey details.

| Province | Facility level* | | | | |
|---|---|---|---|---|---|
| | Primary | Secondary | Tertiary | Quaternary | |
| Harare | 5(2) | – | – | 1(1) | |
| Mashonaland East | 3 | 2 | 1(1) | – | |
| Bulawayo | 6(4) | – | – | 3 | |
| Matabeleland South | 6(1) | 2(1) | 1 | – | |
| **Total** | **19** | **4** | **2** | **5** | **n = 30(10)** |

*Definitions of levels of care:

**Primary**. Clinics, polyclinics, private clinics, mission clinics, council/municipal clinics, rural health centres.

**Secondary**. District hospitals and other public and private health facilities to which patients are referred from primary facilities for more specialist care.

**Tertiary**. Provincial hospitals and other health facilities offering more advanced medical investigations and treatment than available at secondary level.

**Quaternary**. Central hospitals in Harare and Bulawayo.

(n) Facilities also included in participant-observation work (n=10).

consultation with relevant MoHCC directorates. Questions pertained to services and staffing; challenges of managing NCDs and multimorbidity; and preparedness for managing specific NCDs (hypertension, diabetes, chronic respiratory disease, and depression). These diseases were selected as they are the NCDs included in the SARA survey [32] with the exception of depression, which we added to cover mental health. Surveys were administered with facility managers, with input on specific questions by clinicians, pharmacists, and human resources and health information officers.

To capture the management of multimorbidity in practice, we conducted participant-observation and in-depth interviews in 10 facilities purposively selected from the survey to represent different levels of care. Participants were generally nurses or physicians to whom we were referred by facility managers. They had expertise in general medicine, infectious diseases (especially HIV and TB), physical NCDs (including cardiometabolic conditions, oncology, and rheumatology), and mental health (see Table 1). Participant-observation took place over 1–3 days and involved 'shadowing' participants to understand practices and routines related to multimorbidity, during which fieldnotes were taken on encrypted tablets. Interviews were conducted using bespoke interview guides developed from a set of high-level themes derived from qualitative literature and our knowledge of the context. The guides were adapted throughout the study based on emerging findings. Interviews were conducted where possible at participants' places of work during quiet periods. Interviews lasted approximately 1 hour and were audio-recorded unless participants declined, in which case detailed notes were taken instead. No repeat interviews were conducted. To gain a patient perspective, we purposively selected 23 PLWMM from facilities involved in participant-observation to represent differences in age, sex, geographical location, and conditions (S1 Table). We conducted in-depth interviews with all 23 PLWMM to capture illness narratives, understandings of their medical conditions, and experiences accessing care. We asked every third participant to keep a 7-day audio-visual diary on an encrypted smartphone to better understand their routines. Where appropriate (e.g., for elderly PLWMM), family members and carers assisted in the recording of the audio-visual diaries.

To capture how multimorbidity was being engaged with beyond the service delivery level, we conducted in-depth interviews and, where appropriate, participant-observation with professionals working across Zimbabwe's health and academic sectors. This included public health policymakers and practitioners in the MoHCC (n = 7), health informaticians, electronic health record (EHR) experts and/or data managers (n = 3), technical partners and non-governmental organisations (NGOs) (n = 11), and academics/researchers at tertiary education and research institutions (n = 7). Most of the latter were also at the top of their fields in their respective specialities, providing a clinical academic perspective. Most were also lecturers, thus also offering a medical education perspective. Interviews and participant-observation were conducted using the same approach and tools as those used at service delivery level, with some overlap given the porosity of academic and clinical roles. Finally, we held two participatory workshops, one in Bulawayo (including participants from Bulawayo and Matabeleland South, n = 32), and one in Harare (including participants from Harare and Mashonaland East, n = 37). The aim was to collaboratively interpret primary data and co-develop a formative agenda comprised of priority areas, specific recommendations, and focal institutions/departments for taking these forward. Participants were invited to bring their knowledge and experiences to bear on discussions, but also to reflexively consider and potentially challenge their own assumptions through engagement with others' perspectives.

## 2.3. Data analysis

All data were entered into NVivo 14 for analysis. Data was analysed using grounded theory, which is an inductive, theory-generating methodology that produces contextualised, process-oriented theories of social phenomena [33]. Analysis was performed on an ongoing, iterative basis by JD and FM that fed back into the ongoing collection of data including through the refinement of interview guides. Primary data were coded first by mapping and categorising data before identifying key themes emerging from different perspectives. During the collaborative workshops, participants actively engaged in interpreting the preliminary findings presented and played a crucial role in the validation and refinement of emerging themes. Workshop participants further worked together to translate findings into a provisional multimorbidity agenda, which was subsequently refined and finalised during a final round of data analysis and feedback.

## 2.4. Ethical considerations

Ethical approval was sought from the Medical Research Council of Zimbabwe (MRCZ/A/2842), the Joint Research Ethics Committee of the Parirenyatwa Group of Hospitals and University of Zimbabwe Faculty of Medicine and Health Sciences

(386/2021), the City of Harare, and the London School of Hygiene and Tropical Medicine (26469). All participants involved in survey and ethnographic research activities provided written informed consent. Where appropriate, group briefings were conducted at healthcare facilities prior to asking individuals for consent. Before participant-observation, timeframes were agreed upon with participants such that they were always comfortable with the researcher's presence and were free to alter any plans made. Participants were asked for their consent for anonymised quotations to be used, otherwise their words were paraphrased. Before the collaborative workshops, participants who had not taken part in earlier study activities were asked to provide verbal consent for workshop data to be used for research purposes, which was recorded in an Excel spreadsheet.

## 3. Results

Multimorbidity was a new term for many of our respondents. For instance, a familiar scene during interviews was a participant pausing mid-conversation to search for the term: "multi… what do you call it?... bidity". Familiarity with the term was most pronounced in the international research community, HIV programme, and professional healthcare networks. It also continued to grow during the project lifecycle, not least because of our influence. Those already acquainted with the concept generally understood it as the cooccurrence of multiple long-term conditions in one person, with combinations of HIV and NCDs often cited as examples. But exactly what the concept of multimorbidity evoked for participants, the challenges or tensions it brought to the fore, and any reservations they had about its utility depended on their particular positioning in the health system. In the following sections, we describe the key themes that emerged from different 'windows' into multimorbidity: health seeking and delivery; policy and planning; medical training and professionalisation, and health data and research.

### 3.1. Whose multimorbidity is "better"?

"Interviewee: Yes, my [younger] sister is asthmatic and diabetic as well, the one who comes after me is also asthmatic and diabetes, I am asthmatic too… We joke about our illnesses at times comparing who is better than the other, the one with HIV or diabetes plus asthma.

Interviewer: Do you fear that you might have diabetes too?

Interviewee: (laughs loudly) Yes, I am already asthmatic and HIV positive, I can just imagine being told I am now diabetic. Once you have diabetes the next thing is high blood pressure these two go together. Can you imagine all this burden on me? For now, I am okay without knowing, I will ignore the symptoms until I seriously get ill then I know it's time." (Patient_5, Mashonaland East, living with HIV, asthma, hypertension)

"When I had HIV alone, I was confident that chances of living longer are high because I had known some people who were HIV positive for years. However, when I had diabetes, I felt robbed of life." (Patient_14, Bulawayo, living with diabetes & HIV)

No (0/23) PLWMM in our study had previously encountered the term 'multimorbidity'. However, all talked comfortably about the 'multiple' nature of their conditions. As illustrated by the first respondent, they spoke about how different conditions interacted and were associated, the burden they exerted, and which condition combinations were 'better' to have. As the second respondent suggests, PLWMM often foregrounded the differential experience of HIV and NCDs, reflecting the stark difference in the resourcing and organisation of services for these conditions. HIV+ PLWMM were comparatively satisfied with the quality of care for this condition. HIV prevention and treatment was decentralised and delivered by nurses, assisted by community health workers, near their homes, with only complicated cases needing referral/doctor support. Services were free, with consistently available medicine refills (which could also be collected at a community health post), integrated care for opportunistic infections (e.g., TB), and often additional resources and support. In higher-volume

primary clinics especially in Harare, Bulawayo, and provincial centres, services were usually offered in a dedicated HIV clinic, referred to the 'opportunistic infection' or 'OI' clinic with its own staff, often supported by NGO-paid nurses.

The addition of NCDs or mental health conditions, however, dramatically shifted the experience of multimorbidity. Private clinics and certain "bougie" (i.e., high-resource) NGO-run HIV clinics were able to provide integrated, person-centred care for most NCDs, increasingly with an explicit multimorbidity focus. These one-stop-shop models were a common frame of reference among both PLWMM and health workers for what 'good' multimorbidity care for looks like:

"What I learnt at [NGO-run HIV clinic] is that…the patient gets attended to as a whole under one roof, they don't need to go elsewhere for other conditions, tests and collection of results like we do here. We need to upgrade and provide all services under one roof." (HCW_20, sister-in-charge, Bulawayo)

The nurse here was comparing partner-funded HIV-NCD care with the 'standard' of care for most living with NCD-related multimorbidity, which was provided through general clinic outpatient departments (OPD). OPDs were generally busy, overcrowded environments often staffed by only one extremely stressed, overworked and underpaid nurse. Typically this nurse was working with very little in the way of equipment, diagnostics, and medicines needed to perform the basic consultations per the EDLIZ guideline, with particular shortages noted with second-line NCD medications (S2 Table). Unlike for HIV, user fees were required to visit urban clinics (except clients <5 and >65 years old; rural clinics remain free), and all patients also had to pay for their medicines (either at the clinic or, if out of stock, at a private pharmacy). A further major bottleneck is that unlike HIV, for which most tasks have been shifted to nurses, treatment initiation for most NCDs requires seeing a doctor. Given the severe shortage of doctors, this usually meant referral to a hospital clinic. Often, patients went straight to a central hospital, either because they knew that they would not be helped at the primary level or because they were so ill that they required immediate hospital attention. While stable patients were referred to their local clinic OPD for ongoing (self-)management, HIV and NCDs would be managed separately. NCDs also normally required recurring lifelong hospital visits – for quarterly reviews, any changes to medications, for any complications or further conditions. All of this exerted a huge burden on PLWMM. While some felt that they were able to cope with cumulative costs, few were satisfied with the quality of care, and others felt unable to afford to 'keep up'. They frequently made impossible choices about which condition to prioritize, which appointments to attend, or which medicines to buy. Ultimately the toll of health, social, and economic problems often led to secondary complications requiring expensive inpatient care, further conditions, frailty, and diminished coping capacity. Engaging with PLWMM and providers at different levels of care enabled us to sketch a care pathway for long-term conditions (Fig 2) that we used to engage further stakeholder groups.

### 3.1. *Siloes from "the top" and the erosion of primary care*

When talking to policymakers and public health practitioners in Harare and Bulawayo about multimorbidity, one common narrative was that people are now living longer in part because of the successes of HIV care, resulting in a greater burden of NCDs and mental health conditions. Yet, Northern partners preferred to work through ringfenced funding for HIV, TB, and malaria which created 'siloes' within the health system. As one policymaker argued:

"We have had HIV for quite some time, and the survivors of those living with HIV also have other conditions, in that same person you can find malaria, TB and HIV or other non-communicable diseases then it becomes imperative to probably shift. The initial design of these grants was to look at TB, HIV and malaria which created siloes within the health system and the move away from there is taking time". (Policymaker_1, MoHCC, Harare)

Through interviews and participant-observation, we found that these fragmented funding streams had very real consequences in the day-to-day management of the health system that prevented working across different disease areas.

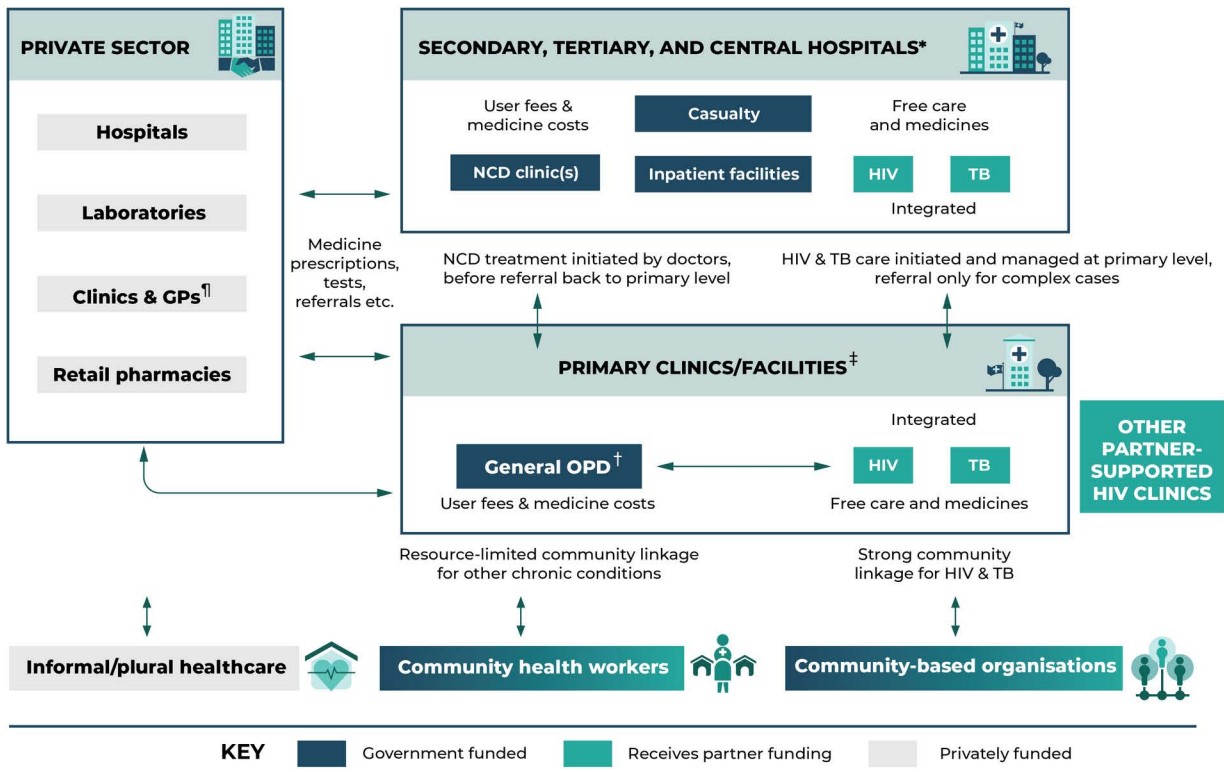

**Fig 2. Illustrative care pathway for long-term conditions and multimorbidity#.** #Considerable heterogeneity exists across provinces and districts, thus this pathway may not be representative of all referral possibilities *Hospitals more specialized at higher levels †Outpatient department ‡Primary care is a nurse-led, doctor-supported service ¶General practitioners.

Certain directorates, notably those relating to HIV programming, were large, well-funded and supported by lower-level implementing personnel at provincial, district, and facility levels. By contrast, directorates with less partner support, for instance those for NCDs and mental health, comprised a handful of individuals, with limited funds to strategize and plan, and far fewer focal persons at lower levels to support implementation. Against this backdrop, an air of competition and protectionism characterized inter-programme relations, likened by one policymaker to fighting over a "carcass" (Policymaker_2, Harare). This worked against collaboration between disease programmes and could result in separate, sometimes contradictory policies and interventions. This was even within areas that had historically been the focus of integration, for instance TB and HIV. Walking with another policymaker through one of the long corridors housing different disease-specific departments, they observed that:

> "The departments of NCDs, HIV, TB and malaria hardly meet to discuss patient issues, but they are all under the same roof which is a challenge already. At the higher level the focus is on the condition and as we go to the lower levels, they see a person and not a condition". (Fieldnote excerpt, Policymaker_3, Harare)

Those at "lower levels" certainly had much to say on the matter. Health workers, facility managers and district/provincial decision-makers, particularly those who had experienced the prior strength of the system in the early postcolonial era, were fierce critics of the current scenario. Perplexed by why multimorbidity was being framed as something new that had arrived once the HIV epidemic had stabilised, one provincial policymaker during a collaborative workshop pointed out that

the country's primary clinics were already designed to be multimorbidity clinics. It was siloes from "the top" that were inhibiting the "already-integrated" nurse from delivering integrated care:

> "Already at the primary healthcare level you get [Bulawayo] clinic, it's a multimorbidity clinic. You go to [Bulawayo] clinic, and it's a multimorbidity clinic…Our priorities should be integration from the top. These siloes come to the already integrated primary care nurse who deals with all the conditions under one roof. Fund those clinics at the right places and let the proper pyramid in primary healthcare work as it used to be." (Policymaker_8, Bulawayo)

Responding to multimorbidity, for her, was a matter of allowing the lower levels of care to do what they were already designed and previously able to do in the 1980s and 1990s before the HIV epidemic.

Integrating services and combatting entrenched 'siloes' was, on the one hand, explicitly written into the National Health Strategy 2021–25 [34]. This was being enacted through a number of promising initiatives including an early-stage pilot of the WHO Package of Essential Non-Communicable Disease (PEN) Programme (designed to decentralise the management of common NCDs), new NCD guidance within the latest HIV Operational and Service Delivery Manual (OSDM) on the integration of HIV-NCD services, and a recent application to the Global Fund to fund medicines for common NCDs among people living with HIV. The latter was hoped to "unlock" previously-inaccessible resources for NCDs: "I believe with integration there is a lot of resources that can be unlocked which can then spill over" (Policymaker_1, Harare). However, many felt that while these were positive steps, they will likely only benefit HIV+ patients and buttress the HIV programme. The underlying issue was that the Northern partners had neither the resources nor the inclination to meaningfully step outside of vertical funding models. Nor was reinvigorating comprehensive primary healthcare felt to be high enough on the government's agenda either. As several suggested, head office was preoccupied with developing specialist capacity at the quaternary level, which would predominantly benefit the country's political and economic elite.

### 3.3. Clinical (sub-)specialism over generalism and public health

This perceived drive towards higher-level, specialist medicine, relative to lower-level 'generalist' care was a key theme when discussing multimorbidity with clinicians and educators. Specialist trajectories were inscribed at multiple levels: from individual pressures to enhance earning potentials and opportunities; to institutional aspirations to become centres of excellence in research, training, and care; to national strategies (e.g., "Education 5.0") to enhance specialist training, biotechnology, and novel drug development. Speaking to one specialist-in-training (Academic_2, Harare), she related that her hospital's mission statement to become a superspecialist centre of excellence by 2030 was a long way off. While specialist training (e.g., internal medicine) was offered at the central teaching hospitals, sub-specialising (e.g., in cardiology, neurology, etc.) required advanced training in South Africa or overseas. Very few were able to afford this, and fewer still would be motivated to return to practice in Zimbabwe especially in the public sector. For those working in the public sector, the ability to practice a (sub-)specialty was severely constrained by available diagnostics, treatments, and patients' resources. Referrals to the private sector – often to one's own private clinic – was often the only available route, but this was not an option in many cases. Such was the staff shortage and resource scarcity that it was impossible to exclusively practice one's speciality. The hospital OPDs and general medical wards were overflowing, many with multimorbidity, in part because of the lack of capacity at lower levels to treat these conditions or prevent them from needing specialist hospital-level attention. As a result, as one sub-specialist put it, "we [sub-specialists] are all general with an interest in one area." (Academic_3, sub-specialist clinical academic, Harare).

If generalism was the reality even for the country's top sub-specialists, it was certainly the case for the majority of physicians working at lower levels of care. As several observed, the problem was that the development of generalist skillsets was not adequately valued or supported. Both classroom learning and rotations were taught around diseases and organ systems – "in siloes", as a lecturer described (Academic_1, sub-specialist physician and lecturer, Harare) – which

was inconducive to learning about multimorbidity. On the one hand, some argued that trainees were given opportunity to consolidate specialist knowledge including about multimorbidity during their internships, which often took place in district hospitals. On the other hand, others argued that expecting students to put it all together 'on the job' was insufficient and reflected a lack of value placed on generalism compared to the advanced training expected of specialists. As one Bulawayo-based general practitioner argued:

> "A good generalist is someone who should be able to take care of the most common and prevalent conditions that burdens the local community that he practices in and should be in a position to be a team leader in the preventive aspect of medicine… Let's try to prevent the conditions from getting worse to require specialist care… But we don't do that, why, because people don't know how important a generalist is, a family physician is, a primary care physician is." (HCW_43, general practitioner, Bulawayo)

That new physicians were 'default GPs', so to speak, was contrasted with high-income settings such as the UK, where post-basic certification was required. In the 1980s, there was a well-subscribed MMed in General Medicine similar to the UK model, but this was discontinued in the 1990s. A more-recently introduced MMed in Family Medicine remained under-subscribed during our research compared to the MMed Internal Medicine (taken by those looking to (sub-)specialise) and the MSc Public Health. We found that training in family medicine was disincentivised by the fact that graduates were, as 'generalists', still not regarded as (or able to charge as) 'specialists': "I think there is then need for a move to formalise some of these professions so that even if they do these kind of training they actually, well I will call them specialized in a way" (Policymaker_4, Matabeleland South). Public health trainees, meanwhile, often ended up transitioning towards either MoHCC head office, NGOs, or research institutions. This not only removed these skillsets from the clinical level but meant that they were unevenly absorbed into the better-funded programme areas.

The training of nurses and other cadres was, on the whole, perceived to be more 'generalist' in orientation. The training of general nurses and village health workers has historically been one of the great strengths of Zimbabwe's health system, with the 'old style nurse' able to not only use the EDLIZ guideline to manage common conditions but to really 'think through' your problems. Standards were felt to have dropped due to resource scarcity, staff attrition, and a lack of trainers and mentors, forcing some older nurses out of retirement. This is at precisely the time when, to quote the provincial policymaker above, the "already integrated primary care nurse" (Policymaker_8, Bulawayo) was being pulled in different directions by disease-specific programmes, disproportionately for HIV, TB, and malaria. Each came with additional evidence-based clinical practice guidelines, training packages, and monitoring and evaluation (M&E) systems. As one policymaker argued, in these programmes there was "no room for other diseases, or usually just a small component" (Policymaker_3, Harare). Such a 'small component' included the recent expansion of OSDM guidance to include hypertension, diabetes, and depression. But there was no in-service training for NCD management beyond the HIV programme, with most nurses relying on their basic training. This was evidenced in our facility survey by few nurses having trained in the last 2 years in diabetes, hypertension, asthma, or depression (S2 Table). Together, falling standards and the fragmentation of training was seen to work against the previously strong generalist capacities underpinning Zimbabwe's nurse-led primary care, so important for managing complex health needs like multimorbidity at lower levels.

### 3.4. Fragmentation and demands of health data

The fragmentation and demands of health data presented another layer of complexity to the multimorbidity challenge. Health informaticians and data experts described the complex data ecosystem that made up the 'back end' of Zimbabwe's health system, which they argued had become progressively more expansive and fragmented in recent years. It comprised patient books/cards and records; disease tally sheets (referred to as the "T-series"); a plethora of single-disease programme-specific registers; and the recent introduction of Electronic Health Records (EHR), intended to phase much

of this data into an integrated electronic format. It also included the MoHCC central health information system, DHIS2 (a widely-used system across Africa), into which routine data sources are fed through structures of upward reporting.

In terms of clinical management and follow-up, tracing patients with multimorbidity was challenging. In theory, patient-level records include all known conditions, including multimorbidity, but this was not always the case. Even at primary clinic level, there were separate books for the OI department and general OPD, often making it hard to keep track of HIV+ patients with NCDs and vice versa, particularly if the patient forgot their book(s). Facility-held records for long-term conditions included HIV-related and TB registers and the 'chronic' register for common NCDs (hypertension, diabetes, asthma, epilepsy, and depression). While facility-held, unlike the patient booklets these registers were more condition-oriented, with the same patient often appearing across registers with no cross-reference, making it challenging to identify patients with multiple conditions. In some facilities, nurses would informally record a patient's other conditions against their name wherever it appeared, so that whoever attends to the patient knows to update for all other conditions:

> "If a patient's clinic number is one on the HPT register and number three on DM register then on their book it will be recorded for example 1HPT and 3DM. It will show their clinic number in each condition they have. The clinic chronic register is all we have. If the patient is on ART his or her OI number will be written on their booklet too". (HCW_9, Acting sister-in-charge, Mashonaland East)

However, once a patient was sorted into a disease-specific register for ongoing management, it was likely that the patient would be fast-tracked to that register in future consultations. In turn, it was less likely that clinicians were thinking about other conditions unless the patient explicitly mentioned them. On the one hand, EHR was felt by many to have promise for ameliorating fragmentation and improving the tracking of patients across facilities and levels of care through unique patient identifiers. Currently, however, EHR is still inconsistently available, often down due to connectivity or electricity issues, and remains exclusive to the public sector. More fundamentally, EHR is still ultimately an e-copy of the paper-based system, with disease 'modules' taking the place of physical registers. This does not actually depart from disease-specific health information and, in fact, may reinforce it. Notably, EHR was initially funded by the HIV programme, whose module is thus more advanced than the 'chronic care' module. There has also been funding provided from the other partner-backed programmes to have their modules developed and updated, much of which has occurred without considering of the overall integration of the system.

Data collected for use beyond the clinical encounter included data for disease surveillance, annual health reporting, M&E for specific programme targets, and policy and resource allocation. This data was, if anything, more single-disease focused. Abstracted from the tally sheets (T-series), programme registers, and other forms, data reported upwards were largely cross-sectional counts of disease conditions rather than data about a single person: "The moment you run an aggregated system certainly there is no way you would expect figures that relate to a single person" (Health_ Infor_3_Policymaker_5, Harare). The data were also heavily biased towards the programme-backed diseases (Fig 3). The better-funded programmes had additional, high-resource M&E infrastructure running alongside MoHCC standard reporting systems. It included standardised programme-specific registers (corresponding to clinical guidelines); numerous indicators; regular visits and back-and-forths on data quality; and M&E officers and analysts working at sub-national level all the way upwards. At the facility level, the heterogeneity of programme documentation meant a considerable paper burden on the nurses. Representatives from one programme would come the one day with demands, another programme the next, and these representatives were felt to have a distinct lack of knowledge of the reality on the ground and the overall impact of the paper burden was having. "It demoralises us!," as one nurse from Matabeleland South exclaimed. Adding to this demoralisation was the fact that, while facilities were encouraged to examine data to identify challenges and solutions, rarely were these efforts reciprocated. This lead to despondency in conveying vital experiential knowledge:

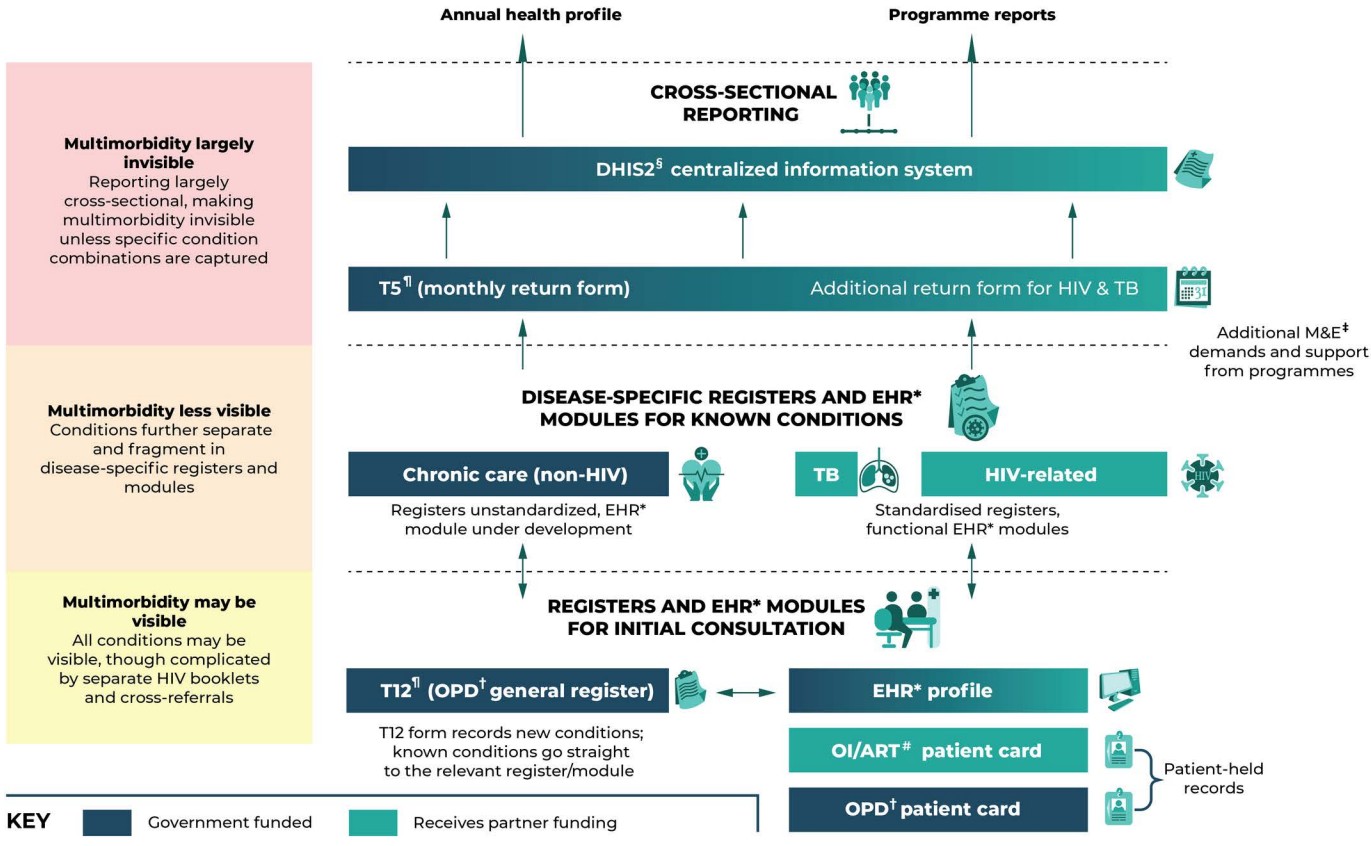

**Fig 3. Data flows for long-term conditions and multimorbidity.** ¶The 'T-series' is a series of tally sheets and forms for capturing and reporting routine data. §DHIS2 is Zimbabwe's centralised health information system, commonly used across sub-Saharan Africa. *Electronic health record. #Opportunistic infection/antiretroviral therapy. †Outpatient department. ‡Monitoring and evaluation.

"We used to sit religiously interrogating the data then we would come up with a list of challenges and proposals. We ended up copying and pasting because nothing was changing. I think these are the things that should be informing policy so that policy changes." (HCW_42_Policymaker_7, District Medical Officer, Matabeleland South)

A further knock-on effect of programme data demands was its detrimental impacts on the reporting of other diseases, for which there were few standardised registers, human resources, or analytic capacities. NCD data were felt to be under-reported and poor quality, with any data entered into DHIS2 going there to "die":

"Nobody reports NCDs! We've lagged for so long we've let things go, even DHIS2 for schizophrenia – that data is there but not consumed, so it dies." (NGO_6, NGO physician, Bulawayo)

On the one hand, there was recognition of the need to improve data for NCDs and, increasingly, multimorbidity. In terms of NCDs, this includes recent efforts to improve M&E infrastructure for NCDs and plans to more regularly conduct the WHO STEPS survey, which includes a component on NCD risk factors (the last of which was conducted in 2006 and is still being used in the National Health Profile [35] and Strategy [34]). There are also diagnostic yields from NCD screening initiatives conducted by HIV implementing partners; routine data available from HIV donor clinics able to support fully integrated NCD care; and new data points in DHIS2 on certain HIV-NCD indicators, corresponding to the expanded

HIV-NCD integration guidance in the HIV guidelines. Finally, there is a growing body of clinical, epidemiological, and social research on multimorbidity in Zimbabwe (our project among them). This research has started to push the horizons beyond what has been a fairly HIV-centric discourse until recently. Of note, however, is that the knowledge architectures within which researchers and programmers have started to engage with multimorbidity remain constrained by the infrastructure already in place for HIV and TB. Thus in our experience, when it comes to questions around where to start – where to integrate from, on what to model our interventions – answers have tended to gravitate towards the known quantities of HIV and TB.

## 4. Discussion

In this study, we pieced together the challenge of multimorbidity from the perspective of multiple actors across Zimbabwe's health system. Doing so when multimorbidity was just emerging on the policy radar allowed us to produce a rich, holistic description to inform priority and agenda-setting while anticipating the possible limitations of elevating it as a specific priority. Multimorbidity almost unanimously resonated with respondents, who framed multimorbidity as a fundamental challenge to the current health system architecture. Multimorbidity was being most actively engaged with from an HIV perspective and by the international research community, with conversations around multimorbidity beyond these groups somewhat less explicit or coordinated.

Our findings build upon a growing body of social research on multimorbidity in Africa. Consistent with findings in Malawi [14], Ghana [36], Ethiopia [15], and South Africa [12,13], we found that for PLWMM, the concept of multimorbidity captured the tremendous health, economic and social burden of navigating the competing demands of different long-term conditions within 'vertical', unevenly-resourced health services. We then showed that the challenges faced by PLWMM reflected more 'upstream' tensions facing Zimbabwe's policymakers and planners. For them, multimorbidity presented as profound managerial disenfranchisement characterised by contestation over the "carcasses" of disease-specific funding – all-too familiar in other accounts of African governments' engagements with 'global health' [37–39]. Such administrative challenges in turn speak to a wider rupture in the biopolitical order from the promises of nation-state-led public healthcare systems following Alma Ata to the bankrupting of such systems and incursion of fragmented donor funding following HIV [40]. With Zimbabwe having made among the greatest strides towards comprehensive primary care [25], the system's downfall and inability to provide basic NCD medicines was experienced as a profound sense of loss of grassroots-level primary care. Further gutting the system was the strain placed on clinical and public health professionals, for whom multimorbidity presented as a series of pressures towards specialised hospital medicine, the private sector, NGOs and research. This corresponds with a weakening of the generalist skillsets that previously made comprehensive care possible at the lower levels [17,41]. Finally, health data, rather than revealing a problem with the status quo, functioned to legitimise it. From a health data perspective, multimorbidity manifested as the 'papering over' of patient complexity through echo chambers of disease-specific data, upward reporting, and feedback, with the 'best' data mediated by analysts in the global North [42]. This made PLWMM difficult to track; imposed a paper burden on nurses that detracted from patient care [17,43]; and made the challenges faced at the care delivery level almost impossible to recognise or respond to higher up. This demoralising situation disempowered nurses and compounded the burden on patients, while reproducing the hypervisibility and prioritisation of already-funded diseases.

Multimorbidity may have been a meaningful and useful concept for most. However, the directionality of multimorbidity discourse, coupled with the perspectives of our respondents, gives reason to pause. Multimorbidity has been promised to lead us towards integrated person-centred care [18], however as critics have pointed out, it has followed a familiar pattern within global health, focused on quantifying and objectifying it such that it can form the basis of new evidence-based interventions [16]. As this has been transposed into a global health problem, multimorbidity has found fertile ground in a 'clashing epidemics' narrative [17,44] that, in Zimbabwe as elsewhere [12], has been readily absorbed within the HIV programme's remit, reflected in growing knowledge of HIV-NCD comorbidity, progress with integrating certain NCDs into

existing HIV structures, and plans to use HIV as a model for NCDs and multimorbidity beyond the HIV+ population [45]. But ethnographic evidence, both in high-income [18] and lower-income settings [12], suggests that expanding or merging vertical programmes will be insufficient to respond to multimorbidity [16,17]. In the UK, Lynch et al. found that multimorbidity has failed to live up to its promise to deliver on person-centred care, partly because the emphasis has been on building bridges between an ever-growing number of (sub-)speciality areas rather than the *repair* or *restoration* of the more comprehensive primary care services that were in place prior to greater specialism [18]. In Zimbabwe, by contrast, the past was a key frame of reference. For many, Zimbabwe was already prepared for multimorbidity before HIV was extracted, given its own clinic, and made into something special. As we showed in prior ethnographic research, Zimbabwe's historically strong, comprehensive primary care system remained a powerful memory and guiding frame [25]. Thus, for many, responding to multimorbidity was as much about restoring or repairing the older system as preparing ourselves for a 'new' emerging pandemic [1].

Accordingly, the formative agenda that emerged from the collaborative workshops was oriented towards restoring the centrality of comprehensive primary healthcare and supporting the 'already-integrated' nurse (S3 Table). The priority areas within the agenda are expansive, but two cross-cutting topics predominated conversations. One was the question of what integrated care meant for existing HIV and 'OI' infrastructure. Such conversations referenced other countries' responses, including South Africa's ICDM model [46–49], the INTE-AFRICA study in Tanzania and Uganda [50], and Malawi's Integrated Chronic Care Clinics [51,52], the latter of which MoHCC representatives had visited recently. Common to such models is the reorganisation of care from an 'HIV vs. all other conditions' model to an 'acute vs. (stable) chronic conditions' model to ensure that all chronic patients, regardless of HIV status, are seen within the same clinic/space. Proposed benefits include: creating a platform for the decentralisation of chronic conditions previously treated at higher levels; more efficient use of clinic space and resources; more patient-centred management plans, guidelines, and training; a harmonised and less burdensome M&E infrastructure; and the destigmatisation of HIV [49,51,53]. To date, these examples remain few in number, early stage, and with challenges of uptake, sustainability, and scale-up [12,54]. The MoHCC and technical partners have recently participated in cross-country discussions exploring possibilities for such a model in this setting [53], and, for the most part, the collaborator group in this study (some of whom took part in that initiative) welcomed such a model. A proviso however was that another parallel silo is not created, which may come from the reification the chronic clinic concept without tailoring it to different settings and levels of care (e.g., in a rural clinic where there was no 'OI' setup to begin with). Insofar as such model could be useful, it would be one focused not on taking chronic disease or multimorbidity *out* but putting HIV back *in*, leveraging resources and infrastructure where needed but only insofar as it restored and strengthened generalist care and the referral pyramid.

A second cross-cutting conversation related to the need for a more enabling 'upstream' health system environment to better embrace change at the service delivery level. It was felt that building a multimorbidity-responsive health system required integrating knowledge and experience across different disease programmes while listening more to ground-level knowledge and expertise. 'Vertical' health financing and fragmented policy structures were recognised as inhibitory to building such an environment, and participants argued the need to be stronger with external partners on using existing pooled financing mechanisms rather than earmarking funds for particular conditions [34]. At the same time, decades of under-resourcing, decapacitation, and attrition have meant a 'culture' that is inconducive to knowledge co-production, collective problem-solving, or innovation. Recent work within health policy and systems research (HPSR) has stressed the importance of investment in learning as a core pillar of health system strengthening [55,56]. Investing in learning means enhancing domestic information systems, research, and analytic capacity to shift away from extractive parallel data systems. Just as important within learning health systems are inclusive deliberative platforms, experiential learning, and 'embedded' research, oriented towards what Gilson et al. have referred to as "collective sensemaking" [57,58]. The concept of health system learning aptly captures the upstream imperatives expressed by our respondents. While radical, it provides an actionable framework for how knowledge, policy, and practice synergise within a multimorbidity-'learning'

health system. Without investment in these more upstream capacities [16], integrated or person-centred care models – the dominant focus of evidence-based interventions – may be undercut by familiar systemic challenges.

Striving towards a 'multimorbidity-learning' health system may seem aspirational, particularly in the context of Zimbabwe's challenging social, political and economic environment. However, the need for bold movements to challenge the status quo is urgent. COVID-19, while underscoring the need to take multimorbidity seriously, also made evident the waning international capacity for supporting lower-income countries beyond securitised public health measures [59]. Donor funding for HIV is expected to reduce drastically by 2030 as part of the transition towards a 'maintenance' model. Against this backdrop, health systems across Africa may soon need to have developed domestic capacity to respond to increasingly complex needs of both those living with and without HIV, while navigating an increasingly securitised global health agenda that may not be responsive to such needs [59]. These developments, while detrimental in one respect, may provide opportunity to further the decolonising health agenda [60,61]. With multimorbidity reigniting calls for more holistic, person-centred approaches in a way the primary care and NCD agendas have largely failed to do [62], it could be a fruitful construct (if framed carefully) for building a shared vision for the future of African health systems. This could be grounded in recognition of not only shared challenges in the current moment but also aspirations towards – or in Zimbabwe's case, back towards – more self-reliant, adaptive, and ultimately more person-centred healthcare systems.

## 5. Conclusions

In this article, we captured and integrated different perspectives on multimorbidity across Zimbabwe's health system. We found that how multimorbidity is framed as a challenge, and by whom, matters. Multimorbidity is hoped to bring about a return to more holistic, upstream, person-centred approaches. But while it may be tempting to frame it as new and pressing, doing so may perpetuate the same challenges it is hoped to overcome [16,17]. Following other ethnographic approaches, we have proposed multimorbidity can point us in a somewhat different direction, focusing on repair and restoration of older systems as an integral part of responding to what is (apparently) new [18]. Of course, it is not simple to turn back the tide, and there is need to consider other countries' experiences with integrating care or otherwise putting systems in place in which HIV is not at the centre. In doing so, there is need to expand our thinking beyond the care delivery level as vertical programming has impacted all aspects of health system functioning [17]. The learning health system, in unsettling the siloes and hierarchies that perpetuate single-disease thinking, may offer a framework for intervening 'systemically' on multimorbidity to (re)build person-centred health systems.

## Supporting information

**S1 Table. Breakdown of participant demographics of people living with multimorbidity (PLWMM).** Provides information about the demographics of PLWMM included in the study, including age, sex, province, and conditions.
(DOCX)

**S2 Table. Availability of medicines, equipment, diagnostics, and training for common NCDs in 30 health facilities.** Provides select quantitative findings from the health facility survey regarding medicines, equipment, and training for NCDs.
(DOCX)

**S3 Table. Priorities and key institutions for responding to multimorbidity in Zimbabwe.** Presents the main priorities and focal institutions for responding to multimorbidity identified during the final collaborative workshop held in Harare 1st December 2023.
(DOCX)

**S1 Checklist. COREQ checklist: Includes a completed COREQ checklist which is commonly used for reporting on qualitative studies.** Domains include: Research team and reflexivity; study design; and analysis and findings. (DOC)

## Acknowledgments

We thank the Zimbabwe Ministry of Health and Child Care (MoHCC), Provincial and District authorities in Harare, Bulawayo, Mashonaland East, and Matabeleland South, the University of Zimbabwe and the National University of Science and Technology for their invaluable guidance, support and collaboration. We express our gratitude to all the participants who took part in the study, whose experiences, insights, expertise and collegiality were invaluable to the study's findings. We also thank all administrative and technical staff who helped to ensure the operational success of the study, including the considerable work that went into organising and implementing the collaborative workshops.

## Author contributions

**Conceptualization:** Justin Dixon, Karen Webb, Clare I. R. Chandler, Rashida A. Ferrand, Chiratidzo E. Ndhlovu.

**Data curation:** Justin Dixon, Fionah Mundoga.

**Formal analysis:** Justin Dixon, Fionah Mundoga.

**Funding acquisition:** Justin Dixon, Karen Webb, Clare I. R. Chandler, Rashida A. Ferrand, Chiratidzo E. Ndhlovu.

**Investigation:** Justin Dixon, Efison Dhodho, Fionah Mundoga, Karen Webb, Pugie Chimberengwa, Trudy Mhlanga, Tatenda Nhapi, Justice Mudavanhu, Lee Nkala, Ronald Nyabereka, Gwati Gwati, Gerald Shambira, Trust Zaranyika, Clare I. R. Chandler, Rashida A. Ferrand.

**Methodology:** Justin Dixon, Efison Dhodho, Fionah Mundoga, Karen Webb, Trudy Mhlanga, Tatenda Nhapi, Clare I. R. Chandler, Rashida A. Ferrand, Chiratidzo E. Ndhlovu.

**Project administration:** Efison Dhodho, Fionah Mundoga, Karen Webb, Pugie Chimberengwa, Trudy Mhlanga, Tatenda Nhapi.

**Resources:** Efison Dhodho, Karen Webb, Pugie Chimberengwa, Trudy Mhlanga, Tatenda Nhapi, Theonevus T. Chinyanga.

**Supervision:** Justin Dixon, Karen Webb, Trudy Mhlanga, Theonevus T. Chinyanga, Clare I. R. Chandler, Rashida A. Ferrand, Chiratidzo E. Ndhlovu.

**Writing – original draft:** Justin Dixon, Fionah Mundoga.

**Writing – review & editing:** Efison Dhodho, Karen Webb, Pugie Chimberengwa, Trudy Mhlanga, Tatenda Nhapi, Theonevus T. Chinyanga, Justice Mudavanhu, Lee Nkala, Ronald Nyabereka, Gwati Gwati, Gerald Shambira, Trust Zaranyika, Clare I. R. Chandler, Rashida A. Ferrand, Chiratidzo E. Ndhlovu.

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
