## [Decision Letter · Decision Letter 0]

17 Dec 2024

PGPH-D-24-01795

Assembling the Challenge of Multimorbidity in Zimbabwe: A Participatory Ethnographic Study

Dear Dr. Dixon,

Thank you for submitting your manuscript to PLOS Global Public Health. After careful consideration, we feel that it has merit but does not fully meet PLOS Global Public Health’s publication criteria as it currently stands. Therefore, we invite you to submit a revised version of the manuscript that addresses the points raised during the review process.

In. addition to the reviewers comments, please address the following comments: 

Some figures on prevalence of NCDs and HIV/AIDS in the study setting would be helpfulPHC facilities are overrepresented in the Facility surveys. Please explain reason for this. In addition, are the selected facilities are public facilities or include private ones also.Is only the research team involved in the developing facility survey tool or developed in consultation with some participants groups? why only "hypertension, diabetes, chronic respiratory disease, and depression" included as NCD co-morbidity?Why the results from the survey are not presented in the manuscript? How these results complement the qualitaive findings? More explanation needed on how themes for the interview guides were developed? any framework or prior studies used to guide developing the themes?

We look forward to receiving your revised manuscript.

Kind regards,

Hassan Haghparast Bidgoli

Academic Editor

Journal Requirements:

**Please only choose the relevant sentences from below**

1. Please clarify all sources of funding (financial or material support) for your study. List the grants (with grant number) or organizations (with url) that supported your study, including funding received from your institution. 

2. State the initials, alongside each funding source, of each author to receive each grant.

3. State what role the funders took in the study. If the funders had no role in your study, please state: “The funders had no role in study design, data collection and analysis, decision to publish, or preparation of the manuscript.”

4. If any authors received a salary from any of your funders, please state which authors and which funders.

3. Please provide an Author Summary. This should appear in your manuscript between the Abstract (if applicable) and the Introduction, and should be 150–200 words long. The aim should be to make your findings accessible to a wide audience that includes both scientists and non-scientists. Sample summaries can be found on our website under Submission Guidelines: 

https://journals.plos.org/globalpublichealth/s/submission-guidelines#loc-parts-of-a-submission

4. Figure 1: please (a) provide a direct link to the base layer of the map (i.e., the country or region border shape) and ensure this is also included in the figure legend; and (b) provide a link to the terms of use / license information for the base layer image or shapefile. We cannot publish proprietary or copyrighted maps (e.g. Google Maps, Mapquest) and the terms of use for your map base layer must be compatible with our CC-BY 4.0 license. 

Reviewers' comments:

Reviewer's Responses to Questions

**Comments to the Author**

1. Does this manuscript meet PLOS Global Public Health’s publication criteria ? Is the manuscript technically sound, and do the data support the conclusions? The manuscript must describe methodologically and ethically rigorous research with conclusions that are appropriately drawn based on the data presented.

Reviewer #1: Yes

Reviewer #2: Partly

2. Has the statistical analysis been performed appropriately and rigorously?

Reviewer #1: N/A

Reviewer #2: N/A

3. Have the authors made all data underlying the findings in their manuscript fully available (please refer to the Data Availability Statement at the start of the manuscript PDF file)?

Reviewer #1: Yes

Reviewer #2: Yes

4. Is the manuscript presented in an intelligible fashion and written in standard English?

Reviewer #1: Yes

Reviewer #2: No

5. Review Comments to the Author

Reviewer #1: This ethnographic study including the perspectives of the main actors involved (from policymakers to health professionals and patients) shows a potential useful framework to design a more person-oriented health care system in Zimbabwe.

There is a small mistake in page 48, legend should be Figure 2

Reviewer #2: ==============================

TITLE:

The title does not convey the purpose of this study. In particular, I do not understand the phrasing “Assembling the Challenge” of Multimorbidity. Maybe this is a phrase that is meaningful in the social sciences field at large – but that phrase has no meaning to me in medical science.

Could the authors choose a title that has more “stand-alone” meaning to readers?

For example, to be parallel with the authors' stated intent: “Multimorbidity and healthcare priorities in Zimbabwe: A participatory ethnographic study”

GENERAL ENGLISH COMMENTS:

Much of the English is written in a circuitous and non-linear fashion. For example, phrases offset with commas could be minimized and written in a style that does not break the reading flow. As currently written, much of the Introduction and Methods come across as disconnected ideas that could be better integrated into a clear and “tight” summary.

For example:

[ABSTRACT, lines 59-60] “Multimorbidity, we found, was new to many…” can simply be written as “We found that multimorbidity was new to many…”

[INTRODUCTION, lines 102-103] “Bosire et al, for instance, demonstrated…” can be simply written as “For instance, Bosire et al. demonstrated…”

There are numerous examples of this nested phrase style throughout the article that would benefit from heavy handed editing.

SCIENCE:

I commend the authors for thinking about the complex and non-trivial aspects of multimorbidity as a construct in guiding (or prioritizing) medical practice. This is a unique endeavor and has potential to offer a model for other regions.

While I find the article to be interesting, I also find there to be very little quantitative or even qualitative data that can be scientifically and rigorously summarized. I am not a qualitative or social science expert, so my ability to remark on these findings is limited. The English is at times difficult to follow and circuitous. I think the paper could benefit from more robust organization. I encourage the authors to remove extraneous terms that are not relevant to their specific hypotheses or goals. The paper could probably be half of its current length with better organization and substantial editing and focus.

6. PLOS authors have the option to publish the peer review history of their article (what does this mean? ). If published, this will include your full peer review and any attached files.

**Do you want your identity to be public for this peer review?** For information about this choice, including consent withdrawal, please see our Privacy Policy .

Reviewer #1: No

Reviewer #2: No

---

## [Editor Report · Decision Letter 1]

19 Mar 2025

Multimorbidity and health system priorities in Zimbabwe: A participatory ethnographic study

PGPH-D-24-01795R1

Dear Dr Dixon,

We are pleased to inform you that your manuscript 'Multimorbidity and health system priorities in Zimbabwe: A participatory ethnographic study' has been provisionally accepted for publication in PLOS Global Public Health.

Best regards,

Hassan Haghparast Bidgoli

Academic Editor

Thank you for addressing the reviewers' concerns. The changes made to the manuscript are satisfactory. There is no further comments from the editor.